# IAEval: A Comprehensive Evaluation of Instance Attribution on Natural Language Understanding

**Peijian Gu**[1][*][†] **Yaozong Shen**[2][*] **Lijie Wang**[2] **Quan Wang**[3][‡] **Hua Wu**[2] **Zhendong Mao**[1]

[1]University of Science and Technology of China, Hefei, China
[2]Baidu Inc., Beijing, China
[3]MOE Key Laboratory of Trustworthy Distributed Computing and Service,
Beijing University of Posts and Telecommunications, Beijing, China
gpj123@mail.ustc.edu.cn   shenyaozong@baidu.com   wangquan@bupt.edu.cn

## Abstract

Instance attribution (IA) aims to identify the training instances leading to the prediction of a test example, helping researchers understand the dataset better and optimize data processing. While many IA methods have been proposed recently, how to evaluate them still remains open. Previous evaluations of IA only focus on one or two dimensions and are not comprehensive. In this work, we introduce IAEval for IA methods, a systematic and comprehensive evaluation scheme covering four significant requirements: sufficiency, completeness, stability and plausibility. We elaborately design novel metrics to measure these requirements for the first time. Three representative IA methods are evaluated under IAEval on four natural language understanding datasets. Extensive experiments confirmed the effectiveness of IAEval and exhibited its ability to provide comprehensive comparison among IA methods. With IAEval, researchers can choose the most suitable IA methods for applications like model debugging.

## 1 Introduction

Along with the fast development and wide application of deep neural networks, a lot of interpretability methods have been proposed to explain the models' predictions, which help people understand the reason behind the models' success and limitations (Ribeiro et al., 2016; Lundberg and Lee, 2017; Linardatos et al., 2020; Hanawa et al., 2021). Such interpretability or attribution methods can be primarily divided into two categories: Feature Attribution (FA) and Instance Attribution (IA). While FA highlights important input features or tokens that support the prediction of the input, IA identifies the training instances leading to the predictions of test examples. The evidences obtained by attribution methods can not only help people understand the model's predictions, but also help researchers debug or optimize models, such as finding problematic training data and improving the data quality.

Plenty of IA methods have been proposed recently (Koh and Liang, 2017; Yeh et al., 2018; Barshan et al., 2020; Pruthi et al., 2020). For a test example, these methods calculate a influence score for each training data, reflecting its influence on the prediction of the test example as shown in the left part of Figure 1. The top ranked training instances are extracted as evidences. While all these methods are potentially chosen to identify evidences, how to evaluate them still remains open. Hanawa et al. (2021) proposed a randomized-test and two heuristics to test whether IA methods satisfy minimal requirements for evidences. Karthikeyan and Søgaard (2021) proposed to evaluate by detecting poisoned training examples. Besides, Pezeshkpour et al. (2021) evaluated IA methods by comparing the prediction difference after removing evidences and a randomized-test. However, all these existing methods consider solely one or two perspectives of evidences such as faithfulness, and they ignored other significant requirements that ideal evidences were expected to meet. It still lacks a holistic evaluation for IA methods.

In order to address the above problems, we propose IAEval, the first systematic and comprehensive evaluation scheme for IA methods. IAEval consists of our elaborately designed metrics quantifying and measuring four important requirements for IA, including sufficiency, completeness, stability and plausibility. Although these requirements have been deeply discussed for FA (DeYoung et al., 2020; Wang et al., 2022), we apply them to IA for the first time by refining their definition and designing metrics for measurement thoroughly. We demonstrate the overall framework of IAEval in the middle part of Figure 1. With IAEval, the evaluation of the IA methods can be conducted in a methodical and holistic manner. Furthermore,

---

[*]Equal Contribution.
[†]Work done during internship at Baidu Inc..
[‡]Corresponding author: Quan Wang.

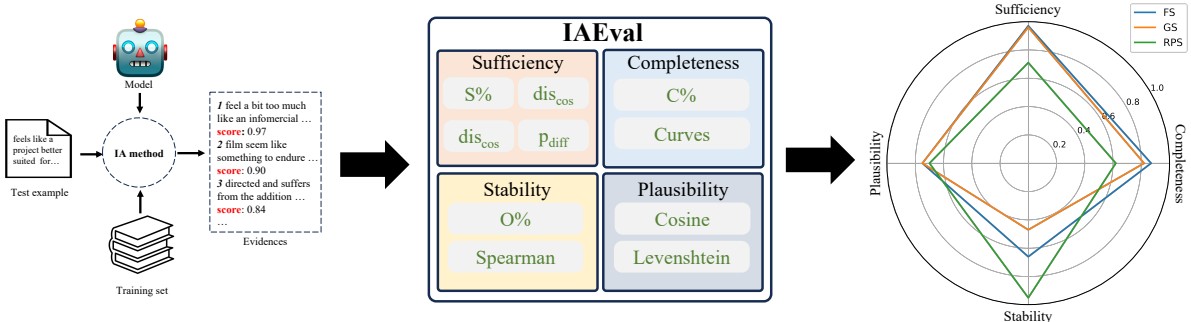

Figure 1: The overall pipeline of our evaluation of IA methods. The left part shows the basic structure of IA methods. They extract evidences according to the influence scores of each training data on a test example. The middle part is our proposed systematic evaluation scheme IAEval. IAEval evaluates IA methods with four sets of metrics covering four significant requirements for IA. The right part is the comparison of the evaluated methods under our IAEval. The comparison is comprehensively conducted on four distinct dimensions.

owing to IAEval's diverse evaluation dimensions, in-depth analyses can be conducted on each IA method. Through such analyses, it is possible to unveil potential drawbacks of a method that may not be detected solely through evaluation from a single perspective (Seeing Section 7.1).

We evaluate IA methods with IAEval on four natural language understanding datasets in two languages: SST-2 (Socher et al., 2013), MNLI (Williams et al., 2018) in English and LCQMC (Liu et al., 2018), Chnsenticorp (Tan and Zhang, 2008) in Chinese. We evaluate three representative IA methods: Feature Similarity (FS), Gradient Similarity (GS) (Charpiat et al., 2019) and Representer Point Selection (RPS) (Yeh et al., 2018). The systematic nature of IAEval enables us make a thorough comparison from four perspectives among these IA methods as in the right part of Figure 1. With IAEval, researchers can refer to its evaluation results and choose better IA methods to acquire higher-quality evidences, helping with specific applications.

Our key contributions are summarized as follows:

- We propose IAEval, the first systematic and comprehensive evaluation scheme for IA methods. IAEval covers four significant requirements including sufficiency, completeness, stability and plausibility. We design novel metrics to measure these four significant requirements for the first time.

- Experiments on four natural language understanding datasets in two languages validate the effectiveness of IAEval. Compared to evaluations solely with a single perspective, IAEval

can provide a comprehensive evaluation and help with unveiling potential drawbacks of a method through multi-perspective analysis.

- With IAEval, researchers can choose the most suitable IA methods for specific applications such as annotation checking or training data reduction.

## 2 Related Work

**Instance attribution.** Recent attribution methods can be primarily divided into two categories: Feature Attribution (FA) and Instance Attribution (IA). In this work, we focus on IA methods which calculate influence scores of each training data for test examples. Influence functions (IF) (Koh and Liang, 2017) approximate the importance of each training data via the products of gradients and the inverse Hessian matrix. Barshan et al. (2020) proposes Relative Influence to mitigate the overlapping problem of IF. Pruthi et al. (2020) measures the importance by tracing the loss change of test example caused by each training data. Similarity-based methods directly measure the training data's importance by calculating the similarity of features (Caruana et al., 1999) or gradients (Charpiat et al., 2019) between training data and test example. Representer Point Selection (RPS) (Yeh et al., 2018) propose to measuring influence score by linearly decomposing the pre-activation prediction of test example with regard to each training instance. With the influences scores, the training data are ranked accordingly and the top ranked instances are identified as evidences.

**Evaluation of attribution methods.** With the advancement of attribution methods, evaluation

of these methods gradually becomes increasingly significant. For FA methods, many datasets with human-annotated evidences have been published for evaluation (DeYoung et al., 2020; Mathew et al., 2021; Camburu et al., 2018; Rajani et al., 2019; Wang et al., 2022). Meanwhile, many studies give their views on the properties that an evidence should satisfy, such as compactness and sufficiency (Lei et al., 2016; DeYoung et al., 2020), comprehensiveness (Kass et al., 1988; Yu et al., 2019), plausibility and faithfulness(Ding and Koehn, 2021; Jacovi and Goldberg, 2020). But for IA methods, there is no manually annotated evaluate datasets, and there are few evaluation metrics. Hanawa et al. (2021) proposes a randomized-test and two heuristics to evaluate the IA methods. Karthikeyan and Søgaard (2021) points out the illness of previous metrics and propose to evaluate by detecting poisoned training examples. Pezeshkpour et al. (2021) analyzes the correlation of different IA methods and makes comparison according to the prediction difference by removing the identified evidences and a randomization test. The above evaluations of IA methods consider solely one or two perspectives. A comprehensive evaluation covering multiple perspectives is an indispensable necessity for IA methods.

## 3 Preliminaries

Firstly, we introduce background knowledge that is used in the later sections. Since most of the existing IA methods are conducted on the classification tasks, we mainly focus on text classification tasks. Given a training dataset $D = \{x_i, y_i\}_{i=1,2,...,n}$ for a task, where $x_i$ is the input sentence or sentence pair and $y_i$ is the label, we train a classification model $\Theta$. We denote the learned representation of $x_i$ as $f_i$ and the classifier linear layer as $\theta$. Given test examples $\{x_t\}_{t=1,2,...,m}$, IA methods quantify the influence score of each training data $x_i$ on the prediction of each testing example $x_t$, denoted as score $I(x_i, x_t)$. For each test example $x_t$, the topk training instances with the greatest influence scores are selected as the evidences $\{\hat{x}_{t,j}\}_{j=1,2,...,k}$.

## 4 IAEval

IAEval covers four significant requirements for IA methods including sufficiency, completeness, stability and plausibility. While these requirements have been discussed for FA (DeYoung et al., 2020; Wang et al., 2022), we conduct the optimization

to tailor them for IA for the first time. We refine their definition and design novel metrics for measurement. We describe in detail each requirement as well as the corresponding metrics as follows.

### 4.1 Sufficiency

**Definition.** The selected evidence are sufficient if they contain enough information for model to make the correct prediction. In other words, the model can make the same prediction only based on evidences as based on all training instances.

**Metrics.** Given m test examples, we use the consistency between the prediction $\hat{y}_t$ supported by the selected evidences and the prediction $\tilde{y}_t$ supported by the full training dataset to measure the sufficiency of the selected evidences. Specially, we train two models, one is based on evidences, and the other is trained on full data. The sufficiency score (denoted as $S\%$) is shown in Equation 1:

$$S\% = \frac{1}{m} \sum_{t=1}^{m} \mathbb{I}(\hat{y}_t = \tilde{y}_t) \times 100\%, \qquad (1)$$

where $\mathbb{I}$ equals to 1 only if $\hat{y}_t = \tilde{y}_t$. The higher the $S\%$ is, the more sufficient the evidences are.

We also calculate the consistency between two prediction probabilities obtained by two models: $\hat{p}_{t_j}$ and $\tilde{p}_{t_j}$. We compute the Euclidean distance and cosine distance between two probabilities as shown in Equation 2 and 3; Meanwhile, we use $p_{diff}$ to measure the difference between them in Equation 4. Lower distance and difference scores imply higher sufficiency.

$$dis_{euc} = \frac{1}{m} \sum_{t=1}^{m} \|\hat{p}_t - \tilde{p}_t\|^2 \qquad (2)$$

$$dis_{cos} = \frac{1}{m} \sum_{t=1}^{m} (1 - cos(\hat{p}_t, \tilde{p}_t)) \qquad (3)$$

$$p_{diff} = \frac{1}{m} \sum_{t=1}^{m} \tilde{p}_{t,\tilde{y}_t} - \hat{p}_{t,\tilde{y}_t} \qquad (4)$$

### 4.2 Completeness

**Definition.** The selected evidences are complete if all the instances that can support test example's prediction are selected, i.e., the remaining training data can not support the prediction of the test example.

**Metrics.** We sort the training instances according to their influence scores, and then divide the training data evenly into N pieces. We then sample the last $n_{sub}$ examples of a data piece as the subset as well as the top $n_{sub}$ examples of the whole training data. As a result, we get N+1 subsets of data representing different influence score rankings. The influences scores in $i$-th subset are higher than those in the $(i + 1)$-th subset. Then for each subset, we train a model and get prediction for test example with the model. We calculate the sufficiency score $S\%$ of each sampling point and report the curve of the $S\%$ with regard to the influence score rankings. To be more concise, we quantify the completeness, denoted as $C\%$, by calculating the maximum difference between the average $S\%$ on both sides of a certain division point in Equation 5:

$$C\% = \max_{1 \le n \le N} \{ \frac{1}{n} \sum_{u=1}^{n} S_u\% - \frac{1}{N+1-n} \sum_{v=n+1}^{N+1} S_v\% \}, \qquad (5)$$

where $S_u\%$ refers to the $S\%$ of the $u$-th subset. Higher $C\%$ implies better completeness.

### 4.3 Stability

**Definition.** An attribution method is stable if it provides similar evidences for similar inputs.

**Metrics.** We use the similarity of evidences for similar test examples as the metric for stability. We calculate the instance-level overlap rate, denoted as $O\%$, between the evidences $\{\hat{x}_{t,j}\}$ of $topN$ most similar test example pairs in Equation 6:

$$O\% = \frac{1}{N} \sum_{(t_1,t_2) \in topN} \frac{|\{\hat{x}_{t_1,j}\} \cap \{\hat{x}_{t_2,j}\}|}{|\{\hat{x}_{t_1,j}\}|} \times 100\%. \qquad (6)$$

Following Pezeshkpour et al. (2021), for the $topN$ most similar test example pairs, we also report the Spearman correlation between the influence rankings over a fixed subset of training set induced from each pair. Higher overlap rate and Spearman correlation imply higher stability.

### 4.4 Plausibility

**Definition.** Plausibility requires that the identified evidences are in line with human cognition and can be understood and accepted by humans.

**Metrics.** The plausibility measures how much the rationales provided by the model align with human-annotated rationales. The golden instance-level evidences are difficult for annotating, as it is unrealistic to annotate an influence score for each training data for each test example, especially with a large training set. To simplify the measure of this evaluation, we propose two metrics for plausibility: semantic similarity and pattern similarity between the selected evidence and the test example. We use the cosine similarity of hidden representations to measure semantic similarity (Reimers and Gurevych, 2019). Meanwhile, we use Levenshtein distance (Levenshtein, 1966) between the dependency trees of the evidence and the test example to measure the pattern similarity (McCoy et al., 2019). Higher cosine similarity and lower Levenshtein distance imply higher plausibility.

## 5 Preparations

### 5.1 Evaluated Methods

In this work, we evaluate three representative IA methods: Feature Similarity, Gradient Similarity (Charpiat et al., 2019) and Representer Point Selection (Yeh et al., 2018). We briefly describe their main ideas and the calculation of influence scores.

**Feature Similarity (FS)** A simple assumption is that higher feature similarity between examples reflects higher influence. We consider the cosine similarity between the learned representation $f_i$ and $f_t$ and the influence score is defined as: $I(x_i, x_t) = cos(f_i, f_t)$.

**Gradient Similarity (GS)** Charpiat et al. (2019) pointed out that the similarity of gradients between two examples reflected how much the change in one would affect the other. We consider the cosine similarity of the gradients $\nabla_\theta \mathcal{L}(x_i, y_i)$ of the classifier linear layer $\theta$. The influence score is defined as: $I(x_i, x_t) = cos(\nabla_\theta \mathcal{L}(x_i, y_i), \nabla_\theta \mathcal{L}(x_t, y_t))$.

**Representer Point Selection (RPS)** Yeh et al. (2018) proved that the pre-activation predictions of test example $\Phi(x_t, \Theta)$ can be decomposed with regard to each training data: $\Phi(x_t, \Theta) = \Sigma_{i=1}^{n} \alpha_i k(x_i, x_t)$. Such decomposition is deemed as the influence of each training data on the test example. The influence score is defined as: $I(x_i, x_t) = \frac{-1}{2\lambda n} \frac{\partial \mathcal{L}(x_i, y_i)}{\partial \Phi(x_t, \Theta)} f_i^T f_t$.

## 5.2 Setup

**Datasets.** We conduct experiments on sentiment analysis dataset SST-2 (Socher et al., 2013) and natural language inference dataset MNLI (Williams et al., 2018). We also experiment on two Chinese datasets: sentiment analysis dataset Chnsenticorp (Tan and Zhang, 2008) and similarity alignment dataset LCQMC (Liu et al., 2018). The statistics of these datasets are listed in Appendix A.

**Models.** We fine-tune the classification models based on RoBERTa-base (Liu et al., 2019) for two English datasets and Chinese RoBERTa-wwm-ext (Cui et al., 2020) for two Chinese datasets. All our experiments are conducted with 5 different seeds and the average results are reported.

**Test Examples.** Pretrained models like RoBERTa (Liu et al., 2019) possess a certain degree of zero/few-shot capability. They are able to predict some examples correctly without or with very few training examples. As a result, the prediction of such simple examples may not depend on any training example and the corresponding evidences are probably meaningless. We avoid to evaluate on these simple data. To filter out them, we randomly sample 100 training examples, fine-tune models accordingly, and make predictions on the dev set. We repeat the procedure for 5 rounds, getting 5 different models and predictions. The examples with 4 and more correct predictions in the 5 rounds are considered simple and neglected. We then randomly sample 100 examples from the remaining data in dev set as our test examples.

## 6 Experiments

With our evaluation scheme IAEval, we make a comprehensive comparison among the aforementioned IA methods from four distinct perspectives. We report the performance of these methods with regard to each important requirements in Table 1.

### 6.1 Main results

**Sufficiency.** We train models with top 100 ranked evidences identified by each method as well as 100 randomly sampled training data and make predictions. We compute our metrics for each models with the model that is trained with the full training data. We report results in Table 1a. All three methods perform remarkably better than random sampling, which indicates our metrics for sufficiency

are effective and can better distinguish IA methods from random test. FS and GS perform fairly close across four datasets, while RPS performs slightly inferior to them on three datasets. For LCQMC, RPS performs much worse on all four metircs.

**Completeness.** For each test example, we extract 10 examples for each subset and train models. We then calculate $S\%$ of every rank point as in Section 4.2. We show the curve of $S\%$ with regard to the rankings in Figure 2. A complete IA method should perfectly separate the whole dataset into two parts. One of them contains all of the evidences while the other one contains few or zero evidences. The perfect completeness curve should look like a 'z' shape, which reaches 100% at the beginning, drops dramatically to 0% at a demarcation point and remains 0% to the end. Figure 2 clearly shows that the curve of FS is more close to perfect. GS has a good curve overall, but there remains flaws at the bottom part. RPS performs worse but still normal on English datasets, and it looks like out of shape on Chinese datasets. The above observations can also be validated through $C\%$ in Table 1a. While FS achieves higher $C\%$ than GS, the score of RPS is significantly lower than the other two methods except on MNLI where all three methods achieve relatively close scores.

**Stability.** We select the top 100 most similar pairs from our test examples according to the cosine similarity of the hidden representations. The cosine similarity is calculated with sentence-BERT (Reimers and Gurevych, 2019) on English datasets and Text2vec[1] on Chinese datasets. We report the overlap rate of their top 1000 ranked evidences in Table 1b. A fixed data subset of size 1000 are sampled from the training set. For each test pair, the Spearman correlation between the influence rankings is calculated on this subset and the average results are reported. We observe that under the Spearman correlation metric, FS and RPS exhibit relatively close performance, while GS performs the worst, indicating its inferior stability compared to the other two methods. Besides, FS and GS have similar overlap rate for all the datasets while RPS achieves an abnormally high score.

**Plausibility.** For each text example, we select top 1000 evidences, and report the average cosine similarity of their representations and Levenshtein

---

[1]https://github.com/shibing624/text2vec

|  | **SST-2** | | | | | **MNLI** | | | | |
|---|---|---|---|---|---|---|---|---|---|---|
| Method | S% | dis$_{euc}$ ↓ | dis$_{cos}$ ↓ | p$_{diff}$ ↓ | C% | S% | dis$_{euc}$ ↓ | dis$_{cos}$ ↓ | p$_{diff}$ ↓ | C% |
| FS | 98.6 | **0.426** | **0.091** | **0.301** | **91.7** | **98.4** | 0.597 | **0.202** | 0.484 | 90.6 |
| GS | **99.8** | 0.432 | 0.094 | 0.302 | 87.7 | 98.2 | **0.593** | 0.206 | **0.481** | 87.6 |
| RPS | 99 | 0.470 | 0.116 | 0.327 | 69.5 | **98.4** | 0.616 | 0.222 | 0.498 | **92.6** |
| random | 28 | 0.701 | 0.312 | 0.495 | N/A | 32.6 | 0.760 | 0.395 | 0.616 | N/A |

|  | **Chnsenticorp** | | | | | **LCQMC** | | | | |
|---|---|---|---|---|---|---|---|---|---|---|
| Method | S% | dis$_{euc}$ ↓ | dis$_{cos}$ ↓ | p$_{diff}$ ↓ | C% | S% | dis$_{euc}$ ↓ | dis$_{cos}$ ↓ | p$_{diff}$ ↓ | C% |
| FS | **100** | **0.094** | **0.004** | 0.053 | **90.6** | 97.8 | **0.110** | **0.007** | **0.045** | **75.0** |
| GS | **100** | 0.104 | 0.005 | **0.052** | 88.6 | **98** | **0.110** | 0.008 | 0.047 | 64.1 |
| RPS | 93 | 0.213 | 0.035 | 0.146 | 58.5 | 65 | 0.478 | 0.206 | 0.323 | 30.4 |
| random | 55.8 | 0.495 | 0.250 | 0.438 | N/A | 47.4 | 0.620 | 0.280 | 0.427 | N/A |

(a) Sufficiency and Completeness

|  | **SST-2** | | **MNLI** | | **Chnsenticorp** | | **LCQMC** | |
|---|---|---|---|---|---|---|---|---|
| Method | Spearman | O% | Spearman | O% | Spearman | O% | Spearman | O% |
| FS | **0.0478** | 10.33 | 0.0292 | 0.31 | 0.0278 | 30.53 | **0.0399** | 1.20 |
| GS | 0.0299 | 10.34 | 0.0265 | 0.04 | 0.0253 | 30.68 | 0.0378 | 1.20 |
| RPS | 0.0400 | **65.95** | **0.0314** | **31.78** | **0.0330** | **57.40** | 0.0322 | **44.03** |

(b) Stability

|  | **SST-2** | | **MNLI** | | **Chnsenti.** | | **LCQMC** | |
|---|---|---|---|---|---|---|---|---|
| Method | Cosine. | Leven. ↓ | Cosine. | Leven. ↓ | Cosine. | Leven. ↓ | Cosine. | Leven. ↓ |
| FS | **0.3045** | 19.15 | **0.1556** | **35.71** | **0.6110** | 85.61 | **0.4060** | **12.53** |
| GS | 0.3036 | 19.14 | 0.1528 | 35.80 | 0.6108 | **85.60** | 0.4059 | 12.54 |
| RPS | 0.1454 | **17.65** | 0.1291 | 37.19 | 0.5750 | 91.24 | 0.3654 | 12.66 |

(c) Plausibility

Table 1: The overall performance of the evaluated IA methods under IAEval. ↓ means the smaller the better. The best results in the corresponding regime are shown in **bold**. a) Sufficiency performance and completeness performance. $C\%$ quantifies the completeness performance in addition to the completeness curves in Figure 2. All the reported results are averaged over 5 different runs. b) Stability performance. c) Plausibility performance.

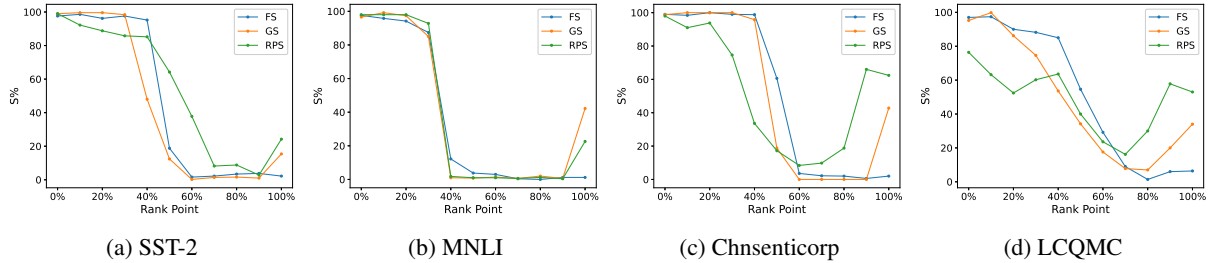

(a) SST-2          (b) MNLI          (c) Chnsenticorp          (d) LCQMC

Figure 2: Completeness performance of evaluated IA methods. We report the S% of each rank point on the four evaluated datasets. All the reported results are averaged over 5 different runs.

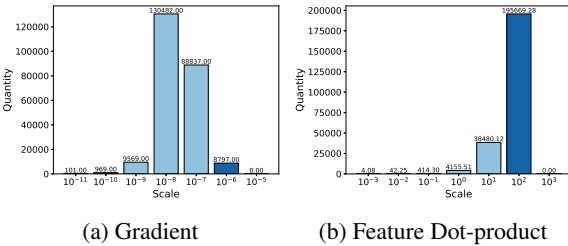

(a) Gradient          (b) Feature Dot-product

Figure 3: The distribution of a) the scale of the gradients of the loss function with respect to the pre-activation logits of each training data. and b) the averaged scale of feature dot-products on LCQMC.

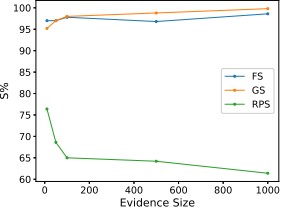

Figure 4: The S% of the evaluated methods based on different evidence sizes on LCQMC.

distance of their dependency trees in Table 1c. The cosine similarity is calculated in the same way as in the stability part. The dependency parsing is done with CoreNLP (Manning et al., 2014) on English datasets and DDParser (Zhang et al., 2020) on Chinese datasets. As shown in Table 1c the performance of FS and GS is relatively close under both metrics, while RPS performs worse than the other two methods except for the Levenshtein distance in SST-2. Such anomaly may be due to the incompleteness of sentences in SST-2 and the dependency is relatively broken compare to other datasets.

## 6.2 Comparison of methods

After the evaluation with IAEval, we claim that FS performs relatively better among the evaluated IA methods. GS is inferior to FS in completeness and stability. RPS performs significantly worse than expectation. The poor performance and abnormal pattern exhibited by RPS on our metrics prompt us to undertake an in-depth analysis, which leads us to a reasonable explanation in Section 7.1 that can account for its underperformance and anomaly.

## 7 Analysis

In this section, we first provide an explanation for the poor performance and anomaly of RPS on LCQMC. To illustrate the robustness of IAEval, we conduct ablation studies with regard to evidence size. Furthermore, we validate the effectiveness of IAEval with an empirical analysis.

### 7.1 Why RPS has a poor performance?

We investigate the reason for the poor performance and abnormally high overlap rate. First let's recall the influence score of RPS: $I(x_i, x_t) = \frac{-1}{2\lambda n} \frac{\partial \mathcal{L}(x_i, y_i)}{\partial \Phi(x_t, \Theta)} f_i^T f_t$. Apart from constants, it consists of two components. One is the gradient of the loss

function with respect to the pre-activation logits of each training data. The other is the feature dot-product of training instances and test example. We discover that the gradient part plays a dominant role in the influence score as shown in Figure 3. The majority of feature dot-products have the largest scale: $10^2$, and they are correlated with the test example. However, only a few amount of gradients have the largest scale: $10^{-6}$, and it is noteworthy that the gradients are independent of the test examples. As a result, when we multiply these two factors, those small amount of training data with high gradients will constantly get a high influence score regardless of the test example. This explains why the identified evidences by RPS have a significant high overlap rate, as shown in Section 6.1. We examined the top 1000 evidences of RPS and discovered that data with gradient of scale $10^{-6}$ dominant the evidences with a proportion of 98%. Such observation verifies our explanation.

Furthermore, we posit high gradient may be attributed to annotation issues. We randomly sampled 100 training data with gradient of scale $10^{-6}$ and examined the annotation. We find that 26% of the sampled data are labeled incorrectly, leading to the poor performance of RPS. We also examined the annotation of 100 training data randomly sampled from whole training data. The annotation error rate is 5%. Such observation inspires us that RPS can be used to extract illness data, helping with annotation checking.

The overall explanation is derived from a comprehensive assessment of performance across four perspectives. Without such a multitude of evaluation angles, explanations for the issue would be partial and the limitations would not be unveiled.

### 7.2 The influence of the size of evidence set

We analyze whether the size of evidence set affects IAEval. The conclusions are consistent on all the datasets and we report the results on LCQMC.

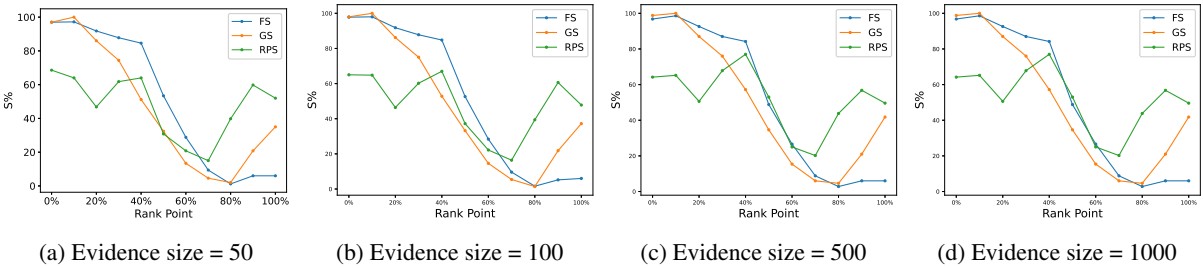

|  (a) Evidence size = 50 | (b) Evidence size = 100 | (c) Evidence size = 500 | (d) Evidence size = 1000 |

Figure 5: Completeness curves of the evaluated methods with different evidence sizes on LCQMC.

| Method | O% | | | | | Cosine Similarity | | | | |
|--------|-----|-----|-----|-----|------|--------|--------|--------|--------|--------|
|  | **10** | **50** | **100** | **500** | **1000** | **10** | **50** | **100** | **500** | **1000** |
| FS | 0.00 | 0.04 | 0.04 | 0.54 | 1.20 | 0.4104 | 0.4063 | 0.4057 | 0.4060 | 0.4060 |
| GS | 0.00 | 0.02 | 0.04 | 0.54 | 1.20 | 0.4094 | 0.4058 | 0.4053 | 0.4058 | 0.4059 |
| RPS | 29.60 | 31.92 | 35.04 | 41.10 | 44.03 | 0.3710 | 0.3718 | 0.3701 | 0.3657 | 0.3654 |

Table 2: The results of stability and plausibility of the evaluated methods with different evidence size on LCQMC.

For sufficiency, we report the $S\%$ of the evaluated methods based on different evidence sizes in Figure 4. The results of FS and GS are relatively constant across different evidence sizes while the RPS meets a performance drop as the evidence size grows. Such drop could be attributed to the growing quantity of the training data with high gradient as in Section 7.1 and the predictions will be biased towards them. As to completeness reported in Figure 5, the completeness curves of the evaluated methods are similar across different evidence sizes. We also report the overlap rate for stability and cosine similarity for plausibility with different evidence sizes in Table 2 and the results are consistent across different evidence sizes.

To summarize, the above observations illustrate that IAEval is consistent and robust with regard to the evidence size.

### 7.3 Validation of effectiveness

To validate the effectiveness of IAEval, we conduct an empirical analysis by simulating training data reduction. We assess the quality of the extracted evidences of each IA method for training data reduction and determine if it is consistent with the conclusion from IAEval. We build a target set by randomly sampling 500 dev examples for each dataset and build a source set by combining the top10 evidences of each target example for each method[2]. The quality of the evidences from each method is measured by the accuracy on the target set, with a model fine-tuned on the corresponding

---

[2]We do not eliminated duplicates and the size of every source set is constantly 5000.

|  | **SST-2** | **MNLI** | **Chnsenti.** | **LCQMC** | **Avg.** |
|--------|-----------|----------|---------------|-----------|----------|
| Full | 95.0 | 86.4 | 93.4 | 86.0 | 90.2 |
| FS | 91.6 | **80.8** | **92.8** | **84.8** | **87.5** |
| GS | **92.6** | 79.2 | 92.2 | 84.6 | 87.1 |
| RPS | 45.0 | 31.8 | 38.0 | 50.4 | 41.3 |

Table 3: The accuracy on the target set. Full represents the fully-trained model. The best results of the evaluated methods are shown in **bold**.

source set. We report the results of the evaluated methods in Table 3, as well as the accuracy of the fully-trained model.

As Table 3 shows, the evidences selected by FS and GS are high-quality and have a competitive performance with the full training data, while RPS performs much worse. Such observation is consistent with the conclusion of IAEval, verifying its effectiveness. The bad performance of RPS may be due to the duplication and relatively bad quality of its extracted evidences as in Section 7.1. Since the high-quality of evidences obtained by FS and GS, we can choose more representative and effective training instances for the given test set, to reduce annotation costs and training time.

## 8 Conclusion

We propose IAEval, the first systematic and comprehensive evaluation scheme for IA methods. We design novel metrics to quantify and measure four important requirements for the first time, including sufficiency, completeness, stability and plausibility. Extensive experiments validate the effctiveness of IAEval. With IAEval, researchers can make comprehensive comparison among IA methods, gain

a thorough understanding of their strengths and weaknesses and choose the most suitable methods for specific applications.

## Limitations

We conduct all the experiments on text/text pair classification tasks while the performance of IA methods on other NLU tasks such as Question Answering are still unstudied. We evaluate three representative IA methods and leave the others for future work because of the high computational cost (influence functions (Koh and Liang, 2017)) or the similarity of methods (dot-product v.s. cosine similarity). Moreover, our experiments are based on the base-size models while the sizes of models are extremely large nowadays. The performance of IA with LLMs may be a possible future research directions.

## Acknowledgements

We would like to thank all the reviewers for their valuable advice to improve this work. This research is supported by National Science Found for Excellent Young Scholars under Grant 62222212 and the General Program of National Natural Science Foundation of China under Grant 62376033.

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

## A Datasets

The statistics of the datasets we evaluate on.

## B Models

We also conduct experiments to show the robustness of IAEval across models in the same language. We report the results of Bert-base-uncased(Devlin

| Dataset | #Train | #Dev | #Test |
|---------|--------|------|-------|
| SST-2 (Socher et al., 2013) | 67,349 | 872 | 1,821 |
| MNLI (Williams et al., 2018) | 392,702 | 9,815 | 9,796 |
| Chnsenticorp (Tan and Zhang, 2008) | 9,600 | 1,200 | 1,200 |
| LCQMC (Liu et al., 2018) | 238,766 | 8,802 | 12,500 |

Table 4: Statistics of the evaluated datasets. We rank the influence scores over the whole training set.

et al., 2018) and RoBERTa-base(Liu et al., 2019) on English; Ernie-3.0(Sun et al., 2021) and RoBERTa-wwm-ext(Cui et al., 2020) on Chinese in Table 5. The consistent results across models indicate the robustness of IAEval across models in the same language.

| | SST-2 | | | | | LCQMC | | | | |
|---|---|---|---|---|---|---|---|---|---|---|
| | **BERT** | | | | | **Ernie** | | | | |
| **Method** | S% | $\text{dis}_{\text{euc}} \downarrow$ | $\text{dis}_{\text{cos}} \downarrow$ | diff $\downarrow$ | C% | S% | $\text{dis}_{\text{euc}} \downarrow$ | $\text{dis}_{\text{cos}} \downarrow$ | diff $\downarrow$ | C% |
| **feature** | 98.4 | 0.291 | 0.042 | 0.206 | 84.8 | 99.6 | 0.219 | 0.022 | 0.146 | 79.8 |
| **gradient** | 99.8 | 0.288 | 0.04 | 0.2 | 57.5 | 98.8 | 0.223 | 0.024 | 0.15 | 65.5 |
| **RPS** | 80.4 | 0.515 | 0.178 | 0.36 | 46.3 | 90.1 | 0.403 | 0.09 | 0.283 | 31.7 |
| | **RoBERTa** | | | | | **RoBERTa** | | | | |
| **Method** | S% | $\text{dis}_{\text{euc}} \downarrow$ | $\text{dis}_{\text{cos}} \downarrow$ | diff $\downarrow$ | C% | S% | $\text{dis}_{\text{euc}} \downarrow$ | $\text{dis}_{\text{cos}} \downarrow$ | diff $\downarrow$ | C% |
| **feature** | 98.6 | 0.426 | 0.091 | 0.301 | 91.7 | 97.8 | 0.11 | 0.007 | 0.045 | 75.0 |
| **gradient** | 99.8 | 0.432 | 0.094 | 0.302 | 87.7 | 98 | 0.11 | 0.008 | 0.047 | 64.1 |
| **RPS** | 99.0 | 0.47 | 0.116 | 0.327 | 69.5 | 65 | 0.478 | 0.206 | 0.323 | 30.4 |

Table 5: Sufficiency and completeness performance of different models in the same language.