# OpenReview forum: "IAEval: A Comprehensive Evaluation of Instance Attribution on Natural Language Understanding"
_EMNLP/2023/Conference — EMNLP 2023 Findings_

### Official Review · Reviewer_wyYh · 2023-08-02

**Soundness:** 3

**Excitement:**

4: Strong: This paper deepens the understanding of some phenomenon or lowers the barriers to an existing research direction.

**Paper Topic And Main Contributions:**

The authors proposed an evaluation framework for Instance Attribution (IA) approaches. The framework covers four perspectives: sufficiently, completeness, stability, and plausibility. The evaluation is on 4 datasets with 3 IA methods.

**Questions For The Authors:**

My concerns are listed in Reasons to Reject

**Reasons To Accept:**

The design of each perspective in IAEval is well-inspired and documented. Multiple datasets and tasks are evaluated in the carefully designed evaluation experiment, and the result consistency (on one model, see comments in Reason to Reject) is maintained across the evaluation.

**Reasons To Reject:**

The IA methods are not model-agnostic, so I would highly recommend the authors show consistent results of different models for the same language (instead of just using one model for one language). The current setup does not fully evaluate and justify the helpfulness of different IA methods, which should be the expected outcome when you are proposing an evaluation metric.
By following the current framing of the paper of proposing an evaluation metric for IA methods, it is necessary to show consistent results of different models. This being said I am able to change the rating based on the rebuttal if the authors have additional results.

As mentioned in the limitation, the experiments are conducted only on classification tasks. The authors made the claim 'all the existing IA methods are conducted on the classification tasks' (line 177), which seems overclaimed and I would suggest the authors to rephrase as long there exist any other IA methods conducted on non-classification tasks.




**Reproducibility:**

4: Could mostly reproduce the results, but there may be some variation because of sample variance or minor variations in their interpretation of the protocol or method.

**Reviewer Confidence:**

4: Quite sure. I tried to check the important points carefully. It's unlikely, though conceivable, that I missed something that should affect my ratings.

---

> ### Author Rebuttal · Authors · 2023-08-28
>
> Thank you very much for your insightful review.
>
>
>
> For concern 1:
>
> The consistency across models is important and we provide the results of our metrics of different models in one language. For English, we report bert-base-uncased and roberta-base. For Chinese, we report ernie-3.0-base-zh and roberta-wwm-ext. We only report experiments on SST-2 and LCQMC due to space limitations. The results are as follows:
>
> Sufficiency and completeness:
>
> |        |  SST2   |         |         |        |      |   LCQMC   |         |         |        |      |
> | :----: | :-----: | :-----: | :-----: | :----: | :--: | :-------: | :-----: | :-----: | :----: | :--: |
> |        |  bert   |         |         |        |      | ernie-3.0 |         |         |        |      |
> | Method |   S%    | dis_euc | dis_cos | p_diff |  C%  |    S%     | dis_euc | dis_cos | p_diff |  C%  |
> |   FS   |  98.4   |  0.291  |  0.042  | 0.206  | 84.8 |   99.6    |  0.219  |  0.022  | 0.146  | 79.8 |
> |   GS   |  99.8   |  0.288  |  0.04   |  0.2   | 57.5 |   98.8    |  0.223  |  0.024  |  0.15  | 65.5 |
> |  RPS   |  80.4   |  0.515  |  0.178  |  0.36  | 46.3 |   90.1    |  0.403  |  0.09   | 0.283  | 31.7 |
> |        | roberta |         |         |        |      |  roberta  |         |         |        |      |
> | Method |   S%    | dis_euc | dis_cos | p_diff |  C%  |    S%     | dis_euc | dis_cos | p_diff |  C%  |
> |   FS   |  98.6   |  0.426  |  0.091  | 0.301  | 91.7 |   97.8    |  0.11   |  0.007  | 0.045  |  75  |
> |   GS   |  99.8   |  0.432  |  0.094  | 0.302  | 87.7 |    98     |  0.11   |  0.008  | 0.047  | 64.1 |
> |  RPS   |   99    |  0.47   |  0.116  | 0.327  | 69.5 |    65     |  0.478  |  0.206  | 0.323  | 30.4 |
>
> Stability:
>
> |        |   SST2   |         |          |         |   LCQMC   |         |          |         |
> | :----: | :------: | :-----: | :------: | :-----: | :-------: | :-----: | :------: | :-----: |
> |        |   bert   |         | roberta  |         | ernie-3.0 |         | roberta  |         |
> | Method | Spearman | Overlap | Spearman | Overlap | Spearman  | Overlap | Spearman | Overlap |
> |   FS   |  0.0537  |  42.57  |  0.0978  |  44.70  |  0.1032   |  11.99  |  0.1716  |  23.09  |
> |   GS   |  0.0378  |  42.78  |  0.0462  |  44.69  |  0.0458   |  12.00  |  0.0940  |  23.10  |
> |  RPS   |  0.0618  |  92.00  |  0.0985  |  96.99  |  0.1304   |  91.2   |  0.1637  |  97.07  |
>
> Plausibility:
>
> |        |  SST2  |       |         |       |   LCQMC   |       |         |       |
> | :----: | :----: | :---: | :-----: | :---: | :-------: | :---: | :-----: | :---: |
> |        |  bert  |       | roberta |       | ernie-3.0 |       | roberta |       |
> | Method | Cosine | Leven | Cosine  | Leven |  Cosine   | Leven | Cosine  | Leven |
> |   FS   | 0.2571 | 18.29 | 0.3045  | 19.15 |  0.7856   | 12.34 |  0.406  | 12.53 |
> |   GS   | 0.2564 | 18.3  | 0.3036  | 19.14 |  0.7856   | 12.34 | 0.4059  | 12.54 |
> |  RPS   | 0.1832 | 18.58 | 0.1454  | 17.65 |  0.7801   | 13.03 | 0.3654  | 12.66 |
>
> As shown in the tables, the performances are consistent across models. Although the specific scores may vary due to different capacity of models, the conclusions are basically consistent.
>
>
>
> For concern 2:
>
> We will rephrase our claim to “most of the the existing IA methods are conducted on the classification tasks” to avoid overclaim.

---

### Official Review · Reviewer_sPvk · 2023-08-05

**Typos Grammar Style And Presentation Improvements:** L237/238
**Soundness:** 4

**Excitement:**

4: Strong: This paper deepens the understanding of some phenomenon or lowers the barriers to an existing research direction.

**Missing References:**

There is another paper in the FA literature that discusses the desired property for evaluation: Jacovi and Goldberg (2020) (https://aclanthology.org/2020.acl-main.386/) which was adopted by an analysis work: Ding and Koehn (2021) (https://aclanthology.org/2021.naacl-main.399.pdf). I found the motivation and design are similar enough to this work so probably worth discussing in the related work.

**Paper Topic And Main Contributions:**

This paper designs a comprehensive evaluation framework for Instance Attribution (IA) methods. The designed evaluation framework is centered around four desired properties discussed in the feature attribution evaluations, namely: sufficiency, completeness, stability, and plausibility. The evaluation was done on four different datasets featuring different text classification tasks and on three Instance Attribution methods, feature similarity (FS), gradient similarity (GS), and Representer Point Selection (RPS).

Results show that FS and GS perform the best, while RPS suffers from some performance issues. Through the analysis, the paper demonstrates that the four properties as outlined in the evaluation framework can provide a comprehensive assessment of IA methods' performance.

**Questions For The Authors:**

Two questions about stability evaluation:

A1. When you select the top N similar examples, did you also control for the prediction to be the same? I think this could eliminate some part of the errors from (1) and (2) I mentioned in the first weakness point.

A2. For (3), do you think it could cause interference to use the similarity of hidden representations both for assigning instance importance (for FS method) and for retrieving top-N instances? From your description, it's not clear to me whether those two similarity metrics are the same (I doubt that it's not, given that FS method does not perform well on the stability metric). Do you think your conclusion would change with an alternative similarity metric?

Then a few other things:

B. For the simple example filtering described in L335-352, how many examples are still left after the filtering?

C. I'm not sure I quite follow the motivation of section 7.3. To me this seems like re-doing the sufficiency evaluation?

**Reasons To Accept:**

1. Each evaluation metric is guided by a clearly motivated goal, which makes the proposal clear and organized.

2. The evaluations are done on four different datasets, including two Chinese datasets, which improves the generality of the conclusions.

3. The experiment and analysis part is thoughtfully designed. I especially find the RPS case study insightful and helpful for the readers to understand how to use those different metrics. The choice to sift out easy test examples for the experiments (discussed in P5) is also a good one.

**Reasons To Reject:**

1. I have some doubts about the proposed stability evaluation. There is no doubt that this property is desirable, but there are many ways this could err. To give a few examples I can think of (1) top N similarity does not really yield similar examples; (2) human label error cause the wrong gradient to be evaluated (for GS); (3) choice of similarity metric to retrieve top N examples interfering with evaluated methods (for FS).

2. The novelty might be deemed as slightly incremental compared to Pezeshkpour et al. (2021), but this is only a minor concern.

**Reproducibility:**

3: Could reproduce the results with some difficulty. The settings of parameters are underspecified or subjectively determined; the training/evaluation data are not widely available.

**Reviewer Confidence:**

3: Pretty sure, but there's a chance I missed something. Although I have a good feel for this area in general, I did not carefully check the paper's details, e.g., the math, experimental design, or novelty.

---

> ### Author Rebuttal · Authors · 2023-08-28
>
> Thank you very much for your insightful review.
>
> For the 3 points in concern 1:
>
> (1) Although the similarity of topN cannot be fully guaranteed, we believe that the current cosine similarity method is prone to capture some similar features. We conduct a statistical analysis and find that all topN pairs actually share the same prediction. Such observation indicates that the similarity metric to retrieve topN similar examples can capture some similar features. Besides, the significant performance drop (e.g., from ﻿77.10% to 0% on Chnsenticorp) between the topN and bottomN similar pairs also indicates the effectiveness of the similarity metric.
>
> (2) The impact of human label error on gradient based methods like GS is inevitable. But since the predictions of the topN similar pair are the same (seeing (1)), it will mitigate such impact to some extent, as the reviewer pointed out in Q.A1. It is impossible to fully eliminate the impact of human label error for gradient based methods such as GS or influence functions, etc.
>
> (3) The similarity metric we use to retrieve topN examples is the same with FS. After eliminating the interference, the conclusion of stability evaluation still holds. Seeing R.A2. for more details.
>
>
>
> For concern 2:
>
> IAEval provides a systematic and comprehensive evaluation scheme from four dimensions. In  Pezeshkpour et al. (2021), the contents related to IAEval mainly consider one dimension and with different designs. Additionally, our analysis and application studies are distinct to theirs.
>
> IAEval evaluates IA methods from four important dimensions: sufficiency, completeness, stability and plausibility. In Pezeshkpour et al. (2021), they mainly considered metrics related to sufficiency while the remaining three important dimensions in IAEval (i.e., completeness, stability, plausibility) are not included in their work. The evaluation of IAEval is more systematic and comprehensive than Pezeshkpour et al. (2021)'s.  Besides, the design of metrics for sufficiency are totally different. They propose remove-n prediction change and a randomized-test while the metrics for sufficiency in IAEval evaluate the prediction consistency between model trained on evidences and the original model. Our metrics are more efficient since we train models only on evidences while they train models on the left training data, much more than the evidences.
>
> The analysis and application studies of our paper is also distinct to Pezeshkpour et al. (2021). They analyze the correlation and computation complexity of IA methods and conduct a artifact-detection application study. In our paper, we analyze the drawback of RPS and conduct application studies on training data reduction.
>
> From the above comparison, it is evident that our evaluation is distinct from Pezeshkpour et al. (2021). The goal of IAEval and the evaluated metrics are novel compared to theirs.
>
>
>
> R.A1:  When we select the top N similar examples, we do not control the predictions to be the same. But after conducting statistical analysis, we found that all pairs actually share the same prediction. Such observation indicates that the similarity metric to retrieve topN similar examples can capture some similar features.
>
>
>
> R.A2: The similarity metric we use to retrieve topN examples is actually the same with FS. We believe such similarity metric could guarantee the similarity of topN pairs to some extend and the discussions in (1) demonstrate that. All the IA methods are evaluated on the same topN examples retrieved with the similarity metric.
>
> However, due to our oversight, we did not realize that it overlaps with the evaluated method FS. As pointed out by the reviewer, it will cause interference for FS. To eliminate such interference, we change the similarity metrics to retrieve top-N instances to the cosine similarity of hidden representations from the pretrained sentence transformers (Sentence-bert and Text2vec). The results are in the following table.
>
> |        | SST2         |         | MNLI     |         |
> | ------ | ------------ | ------- | -------- | ------- |
> | Method | Spearman     | Overlap | Spearman | Overlap |
> | FS     | 0.0478       | 10.33   | 0.292    | 0.31    |
> | GS     | 0.0299       | 10.34   | 0.265    | 0.04    |
> | RPS    | 0.04         | 65.95   | 0.314    | 31.78   |
> |        | Chnsenticorp |         | LCQMC    |         |
> | Method | Spearman     | Overlap | Spearman | Overlap |
> | FS     | 0.0278       | 30.53   | 0.0399   | 1.85    |
> | GS     | 0.0253       | 30.68   | 0.0378   | 1.91    |
> | RPS    | 0.033        | 57.4    | 0.0322   | 35.41   |
>
> As the table shows, although the change of similarity metric to retrieve topN examples affects the metric scores, the conclusions of the stability evaluation still hold.
>
>
>
> R.B: The remaining examples are as follows (remaining/total): LCQMC: 923/﻿8,802; Chnsenticorp: 1116 /1200; SST2: 296/872; MNLI: 6655/9815. Due to the varying complexity of the datasets, the extent of filtering also differs.
>
>
>
> R.C: The motivation of Section 7.3 is to validate the effectiveness of IAEval with an empirical study by simulating training data reduction. The data granularity and purpose of experiments in Section 7.3 is different from sufficiency evaluation. The sufficiency evaluation in Section 6 evaluates the evidences of each test example individually, assessing the performance of each IA method in sufficiency. However, the experiments in Section 7.3 treat evidences of all the test examples as a whole, assessing the performance of each IA method in training data reduction. The former is designed for evaluation within IAEval while the latter is designed for a specific application. Only by assessing the consistency of the results from these two experiments with different purposes can we verify the effectiveness of IAEval. Therefore, the experiments in Section 7.3 are not re-doing sufficiency evaluation.
>
>
>
> Thank you again for pointing out the missing references and typo issues. We will discuss the refered papers in related work.

---

### Official Review · Reviewer_VKZW · 2023-08-06

**Soundness:** 3

**Excitement:**

3: Ambivalent: It has merits (e.g., it reports state-of-the-art results, the idea is nice), but there are key weaknesses (e.g., it describes incremental work), and it can significantly benefit from another round of revision. However, I won't object to accepting it if my co-reviewers champion it.

**Paper Topic And Main Contributions:**

This paper proposes an evaluation framework for instance attribution methods. The authors argue that methods should be evaluated along four axes: sufficiency, completeness, stability and plausibility. The authors evaluate a variety of instance attribution methods (feature similarity, gradient similarity, and representer point selection).

**Questions For The Authors:**

There are some questions embedded in the "Reasons To Reject" above.

**Reasons To Accept:**

- Instance attribution is a hard and important problem, and understanding how to measure progress in the space is an important topic.

**Reasons To Reject:**

- I'm not sure I agree with some of the axes. For example, why does "plausibility" ("requires that the identified evidences are in line with human cognition and can be understood and accepted by humans.) matter? If the goal is to better understand what training instances a model is using to predict a particular test point, why is this related to human understanding at all? In other words, it's completely conceivable to me that a useful and effective instance attribution method has low plausibility, since it's faithful to what the model is actually doing (and the model itself uses features that are uninterpretable to humans)
- Similarly, I'm not sure that I agree with some of the methods used to measure these axes. Going again to plausibility, how is the difference between pattern and cosine similarity supposed to represent if the examples can be understood by humans or not?
- The stated goal is to compare between IA methods, but what makes one IA method better than another? The authors argue that their conclusions "experiments validate the effectiveness of IAEval", but comparing to the random sampling baseline seems like a pretty weak baseline to compare against. In general, I'm not entirely convinced that if you run two IA methods through IAEval, that it really tells you much that'd be useful for comparing these methods, which in general lowers my excitement about this work.

**Reproducibility:**

4: Could mostly reproduce the results, but there may be some variation because of sample variance or minor variations in their interpretation of the protocol or method.

**Reviewer Confidence:**

3: Pretty sure, but there's a chance I missed something. Although I have a good feel for this area in general, I did not carefully check the paper's details, e.g., the math, experimental design, or novelty.

---

> ### Author Rebuttal · Authors · 2023-08-28
>
> Thank you very much for your insightful review.
>
> For concern 1:
>
> Instance attribution (IA) methods are an important branch of interpretability. Since interpretability is for humans, the results of interpretability methods, i.e., the identified evidences in IA, should be in line with human cognition and be understood and accepted by humans. The concept of plausibility has been widely discussed within the interpretability community as in Strout et al (2019) (https://aclanthology.org/W19-48.pdf#page=70), Jacovi and Goldberg (2020) (https://aclanthology.org/2020.acl-main.386/) and Hanawa et al (2021) (https://openreview.net/forum?id=9uvhpyQwzM_), etc. Since plausibility refers to how the explanation is convincing to humans, it is important to evaluate such requirement.
>
> The usefulness and effectiveness of IA methods you mentioned are prone to be reflected in other branches in IAEval such as sufficiency (measuring the prediction consistency between models trained on only evidences and the original model). However, for some domains such as medication or law, the demand for plausibility would be rather high. It is quite necessary to evaluate the plausibility of IA methods for such reason.
>
>
>
> For concern 2:
>
> We believe that when the evidences share similar semantics or have similar structures with the test example, humans can easily recognize these commonalities, thus understanding and accepting the evidences. Conversely, if commonalities cannot be found, humans may feel confused and struggle to understand the evidences. Similarity measurement serves as a way to measure how commonalities can be recognized by humans. Thus, the differences in pattern and cosine (semantic) similarity can reflect differences in plausibility and can represent if the examples can be understood by humans or not.
>
>
>
> For concern 3:
>
> Please note that our goal is not to select the best IA method through comparison; our goal is to propose a comprehensive evaluation scheme that evaluates IA methods from four dimensions including sufficiency, completeness, stability, and plausibility. Additionally, our validation of effectiveness is presented in Section 7.3 rather than in comparing to the random sampling baseline. The scores of random sampling in Table 1(a) serve more like a lower bound for the metrics, helping people grasp the scale of the current metrics. In Section 7.3, we validate the effectiveness of IAEval with an empirical study by simulating the application of training data reduction. In the empirical study, the performance or rankings of the IA methods are consistent with the results of IAEval. The effectiveness of IAEval is thus validated.
>
> In fact, the evaluated metrics in IAEval can align with various demands of practical applications such as sufficiency for training data reduction or plausibility for rationale presentation. IAEval allows users to choose the most suitable IA method for their specific application. What’s more, through experimental results, it can be observed that no method consistently performs well across dimensions. This indicates that there is still room for improvement of IA methods. IAEval can provide valuable guidance for the future design and advancement of IA methods.

---

### Meta-Review · Area_Chair_Vjcu · 2023-09-13

**Recommendation:** 4

**Metareview:**

Quality, clarity, originality and significance

TLDR: The paper proposes a suite of measures -- sufficiency, completeness, stability, and plausibility, to evaluate instance attribution explanations, which would benefit future work on instance attribution methods. An analysis using those is performed on four different classification tasks and three instance attribution methods. The reviewers appreciate the importance of the work and the thoughtful design of measures and experiments. The downside of the work is regarding the plausibility measure, which the reviewers have found both not needed and, most importantly, not correctly designed. Besides, the authors have addressed most reviewers' comments and provided additional results to support their claims further.

Pros:
1. Clarirty/Quality - the design of each metric in IAEval is well motivated and documented (sPvk, wyYh)
2. Significance - instance attribution is an important problem, the work can facilitate future work by providing means to measure progress in the space (VKZW)
3. Quality - the experiment and analysis part is thoughtfully designed, the evaluations are done on four different datasets, improving the generality of observations (sPvk), and the results are consistent across metrics (wyYh)

Cons:
1. Quality  - VKZW raised a concern about plausibility evaluation that was also acknowledged by the other reviewers. The authors should reconsider including plausibility in their suite of measures as 1) the reviewers are debating whether it is always an important property of an explanation and 2) **most importantly, the current way of measuring plausibility does not reflect that an instance is plausible to humans**. In case of missing human annotations, this property should be abandoned. See also the discussion on faithfulness vs. plausiblity in Jacovi et al. where plausibility is defined as "how convincing the interpretation is to humans" where automated measures do not directly measure that.
2. Clarirty -  the authors have addressed most of the concerns of the reviewers and are kindly requested to include all clarifications and additional results in the final version and to tone down their claims that the reviewers found to be overstated.

Jakovi et al. Towards Faithfully Interpretable NLP Systems: How Should We Define and Evaluate Faithfulness?

---

### Decision · Program_Chairs · 2023-10-07

**Decision:**

Accept-Findings

**Comment:**

Quality, clarity, originality and significance

TLDR: The paper proposes a suite of measures -- sufficiency, completeness, stability, and plausibility, to evaluate instance attribution explanations, which would benefit future work on instance attribution methods. An analysis using those is performed on four different classification tasks and three instance attribution methods. The reviewers appreciate the importance of the work and the thoughtful design of measures and experiments. The downside of the work is regarding the plausibility measure, which the reviewers have found both not needed and, most importantly, not correctly designed. Besides, the authors have addressed most reviewers' comments and provided additional results to support their claims further.

Pros:
1. Clarirty/Quality - the design of each metric in IAEval is well motivated and documented (sPvk, wyYh)
2. Significance - instance attribution is an important problem, the work can facilitate future work by providing means to measure progress in the space (VKZW)
3. Quality - the experiment and analysis part is thoughtfully designed, the evaluations are done on four different datasets, improving the generality of observations (sPvk), and the results are consistent across metrics (wyYh)

Cons:
1. Quality  - VKZW raised a concern about plausibility evaluation that was also acknowledged by the other reviewers. The authors should reconsider including plausibility in their suite of measures as 1) the reviewers are debating whether it is always an important property of an explanation and 2) **most importantly, the current way of measuring plausibility does not reflect that an instance is plausible to humans**. In case of missing human annotations, this property should be abandoned. See also the discussion on faithfulness vs. plausiblity in Jacovi et al. where plausibility is defined as "how convincing the interpretation is to humans" where automated measures do not directly measure that.
2. Clarirty -  the authors have addressed most of the concerns of the reviewers and are kindly requested to include all clarifications and additional results in the final version and to tone down their claims that the reviewers found to be overstated.

Jakovi et al. Towards Faithfully Interpretable NLP Systems: How Should We Define and Evaluate Faithfulness?